# Biopsychosocial Aspects in Individuals with Acute and Chronic Rotator Cuff Related Shoulder Pain: Classification Based on a Decision Tree Analysis

**DOI:** 10.3390/diagnostics10110928

**Published:** 2020-11-10

**Authors:** Melina N Haik, Francisco Alburquerque-Sendín, Ricardo A S Fernandes, Danilo H Kamonseki, Lucas A Almeida, Richard E Liebano, Paula R Camargo

**Affiliations:** 1Department of Physical Therapy, Center of Health and Sport Science (CEFID), Universidade do Estado de Santa Catarina, Rua Pascoal Simone 358, Florianópolis, SC 88080-350, Brazil; 2Laboratory of Analysis and Intervention of the Shoulder Complex, Department of Physical Therapy, Universidade Federal de São Carlos, Rodovia Washington Luis km 235, São Carlos, SP 13565-905, Brazil; fisiot.danilo@hotmail.com (D.H.K.); fisio.lucasalmeida@gmail.com (L.A.A.); prcamargo@ufscar.br (P.R.C.); 3Department of Nursing, Pharmacology and Physical Therapy, Universidad de Córdoba, Instituto Maimónides de Investigación Biomédica de Córdoba (IMIBIC), 14004 Córdoba, Spain; falburquerque@uco.es; 4Department of Electrical Engineering, Center for Exact Sciences and Technology (CCET), Universidade Federal de São Carlos, Rodovia Washington Luis km 235, São Carlos, SP 13565-905, Brazil; ricardo.asf@ufscar.br; 5Physiotherapeutic Resources Laboratory, Department of Physical Therapy, Universidade Federal de São Carlos, Rodovia Washington Luis km 235, São Carlos, SP 13565-905, Brazil; liebano@gmail.com

**Keywords:** central sensitization, pain processing, musculoskeletal pain

## Abstract

Biopsychosocial aspects seem to influence the clinical condition of rotator cuff related shoulder pain (RCRSP). However, traditional bivariate and linear analyses may not be sufficiently robust to capture the complex relationships among these aspects. This study determined which biopsychosocial aspects would better classify individuals with acute and chronic RCRSP and described how these aspects interact to create biopsychosocial phenotypes in individuals with acute and chronic RCRSP. Individuals with acute (<six months of pain, *n* = 15) and chronic (≥six months of pain, *n* = 38) RCRSP were included. Sociodemographic data, biological data related to general clinical health status, to shoulder clinical condition and to sensory function, and psychosocial data were collected. Outcomes were compared between groups and a decision tree was used to classify the individuals with acute and chronic RCRSP into different phenotypes hierarchically organized in nodes. Only conditioned pain modulation was different between the groups. However, the tree combined six biopsychosocial aspects to identify seven distinct phenotypes in individuals with RCRSP: three phenotypes of individuals with acute, and four with chronic RCRSP. While the majority of the individuals with chronic RCRSP have no other previous painful complaint besides the shoulder pain and low efficiency of endogenous pain modulation with no signs of biomechanical related pain, individuals with acute RCRSP are more likely to have preserved endogenous pain modulation and unilateral pain with signs of kinesiophobia.

## 1. Introduction

Shoulder pain is a highly prevalent musculoskeletal condition with lifetime prevalence reaching up to 67% across different populations [1]. Rotator cuff related shoulder pain (RCRSP) is a broad and common diagnosis used to encompass disorders associated with rotator cuff problems [2,3]. Although exercise therapy is recommended to be the first-line treatment for individuals with RCRSP [4,5], about 50% of the patients present with persistent pain six to 12 months after the treatment [6,7]. This fact underpins the need for investigating the underlying aspects associated with shoulder pain to better understand the physiopathology of the condition to improve the management of patients with RCRSP.

Biopsychosocial aspects seem to play a role in the clinical presentation of RCRSP [8,9,10]. Among the biological aspects, the extent of tissue damage at the shoulder does not seem to be significantly associated with the intensity of shoulder pain [11,12]. Moreover, strength and range of motion impairments have not been shown to influence patient-reported disability in RCRSP [13]. On the other hand, altered biomechanics at the shoulder seem to play an important role in clinical presentation for a subgroup of individuals with RCRSP [10,14,15]. Additionally, persistent shoulder pain may be associated not only with tissue damage, but also with other biological dysfunctions [8,9,10,16,17].

Somatosensory aspects are also important biological aspects in the pathophysiology of shoulder pain [8,9,10]. Dysfunction in endogenous pain modulation, including abnormal inhibitory and/or facilitatory responses to noxious sensory stimuli [18], distorted perception of somatosensory information on the cortex, such as decreased tactile acuity and poor laterality judgment [19,20], and decreased corticospinal excitability of the rotator cuff were already related with chronicity of RCRSP [17]. However, there is still conflicting evidence about the absence of impairments in pain modulation processes [21,22] versus the presence of central sensitization [8,9,23,24,25] in RCRSP.

Negative emotional and behavioral aspects, such as distress, depression, anxiety, pain catastrophizing, kinesiophobia, fear-avoidance beliefs or low self-efficacy, may influence central pain modulation [18] and contribute to, or even predict, persistent shoulder complaints [26,27,28,29,30,31,32]. However, it is not clear yet which negative or positive social and psychological aspects are more prevalent or lead to a better prediction of persistent complaints in individuals with RCRSP [28,33,34,35,36]. More studies are necessary to understand the contribution of these psychological aspects in RCRSP as evidence is still low [30].

Based on the above, findings about the biopsychosocial aspects in individuals with RCRSP are not consistent among studies. Some aspects seem to influence the chronicity of the condition but may not explain the whole clinical picture of the individuals [8,9,10,18,26,27,28,29,30,32]. The poor predictive capability observed in different biopsychosocial aspects may be due to the multifactorial nature of RCRSP. Understanding how these aspects interact is essential for providing a more tailored treatment. The multivariate and linear analyses traditionally employed to test whether a given correlate and a dependent variable (or group of variables) are associated, while controlling for confounding factors, may not be sufficiently robust to capture the complex interaction among the biopsychosocial aspects in a certain condition [37]. The classification and regression trees are nonparametric statistical procedures that seem to be an alternative approach to capture the nonlinear relationships between multiple heterogenous variables and produce results that can be easily applied in clinical practice [38]. Identifying phenotypes based on biopsychosocial aspects in individuals with RCRSP might contribute to a better understanding of the complexity of RCRSP and guide treatment selection to individuals with acute and chronic RCRSP. The aim of this study was to determine biopsychosocial aspects related to acute and chronic RCRSP and describe how these aspects interact in a nonlinear manner to created biopsychosocial phenotypes in individuals with acute and chronic RCRSP. 

## 2. Materials and Methods 

### 2.1. Study Participants

This is a cross-sectional and blinded study. Recruitment and data collection were performed between November 2017 and September 2018 at the Laboratory of Analysis and Intervention of the Shoulder Complex at Universidade Federal de São Carlos, Brazil. Participants were recruited from the local community, the multidisciplinary clinic of the university, the university campus via email, flyers and social media advertising, and through personal contacts of the investigators. Individuals aged between 18 and 80 years old were eligible to participate if they had rotator cuff related shoulder pain (RCRSP). Inclusion criteria were pain over the deltoid and/or upper arm region for more than 4 weeks, pain associated with arm movement, and familiar pain reproduced with loading or resisted testing during abduction or external rotation of the arm [2]. Participants were classified with acute RCRSP if duration of pain was less than 6 months or with chronic RCRSP if duration was equal or greater than 6 months, as recommended by the International Association for the Study of Pain for research purposes [39].

Potential individuals were excluded for the following: body mass index above 28 kg/m^2^ [40], history of fracture or surgery at the shoulder or cervicothoracic spine [29], signs of adhesive capsulitis (loss of more than 50% of passive range of motion in any direction) [2,41] or instability [42], cervical radiculopathy radiating to shoulder [2,43], physical therapy treatment or corticosteroid injection within 6 months prior to evaluation, history of cancer, neurologic, systemic, rheumatic or vascular disorder [2,44,45], cognitive impairments [45], use of psychiatric medication [46,47] or previous diagnosis associated with central sensitivity syndromes, as determined by part B of the Central Sensitization Inventory [48]. In central sensitivity syndromes, no specific organic cause can be found, and central sensitization is the root cause of the disorders [48]. Therefore, individuals with a well-known central sensitivity syndromes diagnosis, such as irritable bowel syndrome, chronic fatigue or fibromyalgia, were excluded to avoid bias of having pain modulation impairments arising from central sensitivity syndrome within biopsychosocial aspects of RCRSP [49,50]. 

All participants gave their informed consent for inclusion before they participated the study. The study was conducted according to the Declaration of Helsinki, and the protocol was approved by the Human Research Ethics Committee of the Federal University of São Carlos (CAAE 71447317.6.0000.5504). 

### 2.2. Procedures

A physical therapist with 8 years of experience in the clinical setting assessed the inclusion and exclusion criteria and collected the sociodemographic and biological aspects related to general clinical health status and to shoulder clinical condition. A second assessor, with more than 10 years of experience in the clinical setting and blinded to the duration of shoulder pain of the participant, collected biological aspects related to sensory function and psychosocial aspects. The whole evaluation procedure lasted about 90 min. All participants were advised to not consume coffee or alcohol up to 24 h before assessments [51] and to avoid using analgesic medications during the previous 72 h [52]. An overview of the different methods used with their specific outcomes can be found in the Appendix A.

### 2.3. Sociodemographic Aspects

Age, sex, education level, marital status, and dominance were collected. Information on history of employment status, type of occupational demand, and presence of upper limb repetitive movement were also collected. Lifestyle aspects included physical activity frequency (days per week) and smoking status (yes/no) based on self-report.

### 2.4. Biological Aspects Related to General Clinical Health Status

Physical aspects were collected through an interview and physical exam. Outcomes potentially related to chronic pain [50], such as presence of any kind of hypersensitivity (no/yes: light, touch, noise, mechanical pressure, temperature, chemistry) [50], self-reported presence of any other frequent symptoms (fatigue, difficulty to concentrate, sleep disturbance, swollen feeling, tingling, or numbness) [50], presence of any other pain previous to shoulder pain (yes/no) and the pain intensity of the previous most painful complaint in addition to the shoulder pain (0–10 on Numerical Pain Rating Scale, NPRS) [50] were collected. The 11-point NPRS is a reliable and valid scale to assess pain in different musculoskeletal conditions with intraclass correlation coefficient (ICC) ranging from 0.93 to 0.99 [53,54,55]. When no other musculoskeletal pain was concomitantly present, the intensity was registered as zero. 

Part A of the Brazilian version of the Central Sensitization Inventory (CSI) was used to measure central sensitization degree [56]. It is a patient-reported outcome measure with 25 items that assess somatic and emotional symptoms mediated by the central nervous system. Total score ranges from 0 to 100, where the higher the score, the higher the central sensitization. Reliability of CSI showed Cronbach’s alpha of 0.91 [56].

### 2.5. Biological Aspects Related to Shoulder Clinical Condition 

Self-reported aspects related to shoulder clinical condition were unilateral or bilateral pain, duration of symptoms (months), and pain intensity during arm elevation (0–10 on the NPRS) [57]. Abduction and external rotation range of motion (ROM) were collected using a digital inclinometer (Acumar, Model ACU 360, Lafayette Instrument Company). For abduction, the individual was seated with the trunk upright and thumb pointed up toward the ceiling [58]. For external rotation, the individual was supine, with hips and knees flexed, tested arm supported on the table in 90 degrees of abduction, elbow flexed to 90 degrees and wrist in neutral. A towel roll was placed under the arm to ensure neutral horizontal positioning [58]. In both movements, the individual was asked to indicate when the first discomfort was felt (angular onset of pain) and also when the discomfort stopped (or at maximal ROM, angular offset of pain). The presence of scapular dyskinesis was assessed using the dynamic scapular dyskinesis test [59]. It is based on the visual observation of the medial, inferior, and superior scapula borders during 5 trials of arm elevation in the sagittal and scapular planes. Scapular dyskinesis was considered as present (“yes”) when winging, any border prominence or lack of a smooth coordinated movement was observed, and as absent (“no”) when no alteration was found [59,60]. Scapular assistance test was used to assess the influence of scapular motion on shoulder pain [61]. First, the individual was asked to actively elevate the arm and rate the pain felt while performing the movement on the NPRS (0–10). Then, the assessor stood behind the individual and assisted the scapular movement while the individual again elevated the arm. Assessor assisted the scapular’s upward rotation by pushing upward and laterally with one hand over the inferior angle of the scapula, and posterior tilt by pulling backward with the other hand on the superior aspect of the scapula [61]. The individual again rated the pain during the movement on the NPRS. A reduction of 2 or more points during assisted elevation, compared to non-assisted elevation, was considered a positive test. Inter-rater percentage of agreement and kappa coefficient are 79% and 0.40 for scapular dyskinesis [59] and 77% to 91% and 0.53 to 0.62 for the scapular assistance test [61].

The following special tests for RCRSP were also performed: Neer [62], Jobe [63], Hawkins-Kennedy [64], resisted shoulder external rotation [65], Speed [66], Gerber, Yocum, cross-body adduction [66], acromion clavicular pressure, and belly press. The total number of positive tests was used for analysis. All physical outcomes were selected because they are commonly used in clinical and research practice and seem to be associated with RCRSP [2,3,4].

The Brazilian version of the Disabilities of the Arm, Shoulder and Hand Questionnaire (DASH) was used to measure disability of the upper limbs [67]. The DASH is a patient-reported outcome with 30 items assessing self-ability to perform daily activities and severity of symptoms. Total scores range from 0 to 100, where higher scores indicate higher disability. This is a valid and reliable patient-reported outcome to assess individuals with upper limb disorders [67].

### 2.6. Biological Aspects Related to Sensory Function

Quantitative sensory testing (QST) was used to assess sensitivity to a range of stimuli accordingly to the German Research Network on Neuropathic Pain Protocol [68], based on the modified order of testing suggested by Grone et al. [69] to avoid mechanical hyperalgesia after thermal stimulus. The following order of test was performed: 1. two-point discrimination threshold, 2. pressure pain threshold, 3. temporal summation, and 4. conditioned pain modulation. A fifth sensory test, the left/right judgement task, was also performed and the moment for its application was randomly decided by a draw using sealed opaque envelopes because this test does not influence sensitivity [70]. A 5 min rest was provided between each sensory testing. 

#### 2.6.1. Two-Point Discrimination Threshold (TPDT)

To assess tactile acuity, the TPDT was determined by using a mechanical digital caliper with a precision of 1 mm (Digimex, 150 mm). The caliper was applied with an initial distance of 60 mm between the two points and the distance was gradually decreased or increased according to the correct or wrong response, respectively. Single points were applied about every six applications as sham trials to minimize bias [71,72]. For each three reversals (turn point between an ascending and a descending staircase of applications), changes in distance were reduced following the 4-2-1 stepping algorithm described by Wikstrom et al. [72], which demonstrated an intra and inter-rater ICC between 0.76 and 0.93 [72]. The TPDT was the smallest distance in millimeters between two points perceived by the individual after three consecutive reversals at the 1 mm increment [72]. The smaller the TPDT the better the tactile acuity [71]. Anterior (TPDT-anterior) and posterior (TDPT-posterior) shoulder regions of the affected (or most affected) side were assessed over vertical lines drawn from the anterior and posterior edges of acromion, respectively, towards the elbow [71]. Participants were assessed in the seated position, with forearms resting comfortably over a pillow and eyes blindfolded (Figure 1a). Normative values for TPDT have been reported to be between 40 and 45 mm in healthy individuals [71]. 

#### 2.6.2. Left/Right Judgment Task (LRJT)

The Recognise^TM^ shoulder application (Noi Group, Adelaide, Australia) was used to measure body image performance. Participants were asked to judge the laterality of 50 upper limb images, which were shown in different positions in a random order, as accurately and as quickly as possible [70,71]. Familiarization with the application was done using the Recognise^TM^ foot application (Noi Group, Adelaide, Australia) with 5 images of the foot in different positions. Accuracy (LRJT-accuracy) and response time (LRJT-time) were recorded. Accuracy was defined as the percentage of images correctly judged, and response time as the time in seconds dispended to decide whether the picture showed a right or left shoulder [70]. The tool is valid and reliable to measure body image of the shoulder with normative values ranging between 94 and 95% for accuracy and around 1.3 s for response time [71].

#### 2.6.3. Pressure Pain Threshold (PPT)

The PPT, defined as the minimum amount of pressure that provoked the first onset of pain, was measured with a pressure algometer (Somedic AB, Farsta, Sweden), with a 1 cm^2^ probe tip and 40 kPa/s of application rate, as previously described [22]. Local thresholds were measured at a bone surface (PPT-acromion) and a muscle belly region (PPT-deltoid) of the affected (or most affected) side, as well as at a remote region (PPT-tibialis anterior). The side for tibialis anterior assessment was defined by a computer-generated randomization list (randomization.com). The nonpainful side was used as the remote region for those with pain at the lower limb. The order for testing regions was also defined through a computer-generated randomization list stored in sealed opaque envelops and revealed immediately before the start of testing. Three repetitions were performed at each site with 30 s of interval between them, and the average was used for further analysis. Familiarization with the procedure was done at the lateral epicondyle. Participants were seated in a comfortable position and instructed to actively maintain the trunk stable during pressure application (Figure 1b). Reliability of PPT has been shown with the intraclass correlated coefficient, ranging from 0.82 to 0.97 [73].

#### 2.6.4. Temporal Summation (TS)

As a QST dynamic assessment, temporal summation testing assesses one of the pain modulation mechanisms that occurs in the second order sensory neurons in the spinal cord [74]. With the repetition of a nociceptive stimulus at the same region, induced pain seems to enhance from the first to the last stimulus under certain chronic pain disorders [75]. Temporal summation was assessed at the same regions and participant positioning assessed by the PPT (TS-acromion, TS-deltoid and TS-tibialis anterior), with a 2 min interval between testing regions. The sequential stimulation consisted of 10 pressure stimuli at the previously determined PPT level, with a one-second interstimulus interval [76]. The rate of increment of the pressure was about 40 kPa/s for each stimulus [76]. After the first and tenth stimuli, the participant was asked to rate the pain intensity on the NPRS. The outcome measure for temporal summation was calculated by subtracting the pain intensity of the first from the tenth stimulus [76].

#### 2.6.5. Conditioned Pain Modulation

Another QST dynamic assessment is the capacity of the central system to inhibit pain through the conditioned pain modulation (CPM) model, a psychophysical paradigm in which a conditioning stimulus is used to affect a test stimulus reflecting the efficiency of endogenous descending inhibition (or CPM efficiency) [77]. The CPM efficiency was assessed through the combination of the painful conditioning stimulus from the cold pressor test with the test stimulus from the PPT assessment, since this combination induces the most reliable and powerful CPM effect [78,79]. For the cold pressor test, the hand of the contralateral side was immersed up to the wrist into a stirred cold water (4 ± 1 °C) for 2 min. PPT at the acromion of the affected (or most affected) side was tested at baseline, during (after 30, 60 and 90 s of immersion) and after (immediately following withdrawal, and 30 and 60 s post-withdrawal from the water) the conditioning stimulus. The average of the three measures during and post-cold pressor test were calculated separately and used for analysis. CPM was expressed as percentage changes of the PPT-acromion from baseline to during the cold pressor test (CPM during cold pressor test) and from baseline to after the cold pressor test (CPM post-cold pressor test) [77]. 

### 2.7. Psychosocial Aspects

Psychosocial aspects were assessed using the patient-reported outcome measurements described below. The order of application of the questionnaires was randomized by a draw using opaque envelopes immediately before the testing session. 

The Brazilian version of Fear-Avoidance Beliefs Questionnaire (FABQ-Br) [80] and Tampa Scale for Kinesiophobia (TSK) [81] were used to measure fear-avoidance and kinesiophobia behaviors, respectively. The FABQ is a 16 item questionnaire divided into 2 domains that assess fear and beliefs about how physical activity (FABQ-PA) and work (FABQ-W) impact self-perception of pain [82]. Total scores range from 0 to 96 points. TSK is a 17-item scale that measures fear related to injury or re-injury caused by movement [83]. Total scores range from 0 to 68. In both scales, the higher the score, the higher the fear-avoidance or kinesiophobia behaviors, respectively. Test-retest reliability showed an intraclass correlation coefficient (ICC) of 0.94 for FABQ-PA, 0.82 for FABQ-W and 0.82 for TSK [84]. Both questionnaires were used because although fear-avoidance and kinesiophobia are often used interchangeably, they encompass distinct conceptual definitions [85]. Kinesiophobia refers to the extreme fear of movement or physical activity that results from a pain vulnerability awareness or fear of reinjury [85]. Fear-avoidance refers to the extreme behavior of avoiding an identifiable threat with physiological, cognitive and behavioral responses, which leads to the maintenance or exacerbation of fear [85].

The Brazilian version of the Pain Catastrophizing Scale (PCS) was used to measure catastrophic thoughts about pain [86]. It is a 13-item scale divided into 3 domains: rumination, magnification, and helplessness. Total scores range from 0 to 52. The higher the score, the higher the catastrophizing of the individual. Test-retest reliability of this version of PCS showed an ICC of 0.88 [87].

The Chronic Pain Self-Efficacy Scale (CPSS) translated and adapted to Brazilian-Portuguese was applied to measure individual self-efficacy and self-ability to cope with pain and symptom consequences [88]. The scale is composed of 22 items divided into 3 domains: pain self-efficacy, physical activity self-efficacy and symptom self-efficacy. Total scores range from 200 to 2000. The higher the score, the higher is the self-efficacy belief. Internal consistency of the scale showed Cronbach’s alpha of 0.94 for all items [88]. 

The Depression, Anxiety and Stress Scale-21 (DASS-21) is a 21-item scale divided into 3 domains and was used to assess negative emotional aspects of each domain: depression, anxiety, and stress. Total scores range from 0 to 63, where the higher the score, the higher psychological impairment. The Brazilian version of DASS-21 is reliable and valid [89]. Internal consistency of the scale showed Cronbach’s alpha of 0.92 for depression, 0.90 for stress, and 0.86 for anxiety [89].

The Brazilian version of EuroQoL instrument (EQ-5 D) was used to measure quality of life [90]. It is a generic instrument that measures health status in 5 domains: mobility, self-care, usual activities, pain/discomfort, and anxiety/depression. Health state is converted to a code and finally to an index culturally adapted to Brazil that ranges from 0 to 1, where the higher the index, the better the quality of life [90]. Test-retest reliability of the EQ-5 D showed an ICC of 0.58 to 0.89 [91].

### 2.8. Statistical Analysis

Continuous data are presented as mean (standard deviation) or median (interquartile range) according to the Shapiro–Wilk test of normality. Categorical variables are presented as counts (percentages). Between-group comparisons were performed using Student’s *t*-test for independent samples or Mann–Whitney U test for continuous data and Pearson’s chi-square test or Fisher’s exact test for categorical data. 

Decision trees are tools based on divide-and-conquer strategies as a form of learning by induction [92]. This machine learning technique uses a tree structure to classify patterns in datasets, which are hierarchically organized in a set of interconnected nodes. Thus, the nodes considered as leaves classify the instances (inputs) in accordance with their associated label (output). In this sense, the hierarchical structure allows the confirmation of the classification of RCRSP as acute or chronic.

This study used decision trees of the type J48, where the Weka open-source software was employed. In this sense, all the outcomes assessed in this study were used as inputs, with the acute or chronic RCRSP as the predicted variable (output). Thus, the decision tree had its confidence factor parameter adjusted to 0.95. The decision trees were trained and validated through a leave-one-out cross-validation process. For this cross-validation process, one sample of the dataset is used to validate the decision tree, while the remaining data are used for training. This procedure is repeated consecutively until all samples are used in the validation process. The cross-validation is recommended for use in health-related studies, where a complex relationship between the reality of illness and human physiology causes heterogenous and inconsistent data, and where the number of samples are commonly low, in order to surpass the overfitting issue [93,94,95]. Number of leaves was not previously limited.

## 3. Results

### 3.1. Study Population

One hundred and ninety-six potential individuals were initially recruited. One hundred and forty-three individuals were excluded, and reasons are presented in the flowchart of the study (Figure 2). Fifteen individuals (28.3%) had acute RCRSP and 38 (71.7%) had chronic RCRSP. 

### 3.2. Population characteristics

Table 1 displays the sociodemographic and clinical aspects of the participants. Males and young adults were the majority of the individuals included in the study. There was no significant difference between individuals with acute and chronic RCRSP for any of the outcomes, other than the duration of symptoms. 

Table 2 displays the biological aspects related to sensory function and psychosocial aspects of the participants. Individuals with acute RCRSP showed CPM during the cold pressor test 39.4% higher than those with chronic RCRSP (*p* < 0.05). No other significant difference between individuals with acute and chronic RCRSP was observed in Table 2.

### 3.3. Clinical Decision Tree

Classifications of the individuals with acute and chronic RCRSP resulted in a decision tree with 98.08% accuracy (Figure 3). Analyzing the resulting decision tree, it is possible to notice that the cutoff points adjusted during training are quite accurate, since the classification leaves (highlighted in blue) mostly separate one class from the other. Only the classification leaf resulting from bilateral shoulder pain equals “yes” presented a misclassification in which six the seven individuals have chronic RCRSP (85.71%) and one individual has acute RCRSP (14.29%). Therefore, it can be said that this leaf is better suited to classify individuals who have chronic RCRSP and therefore individuals with acute RCRSP classified on that leaf can be considered as errors. However, it is important to note that this one individual misclassification of acute and bilateral pain was previously and correctly identified as an acute case with less than seven points of pain in other regions and CPM during the cold pressor test higher than 75.8%.

Overall, the classification tree presented four levels combining six biopsychosocial aspects (pain intensity of the previous most painful complaint in addition to shoulder pain, CPM during cold pressor test, scapular assistance test, angular onset of pain during external rotation, bilateral shoulder pain and TSK) to distinguish between individuals with acute and chronic RCRSP (Figure 3). Then, the organization of the tree identified seven main combinations between those aspects representing seven phenotypes of individuals: three phenotypes of individuals with acute RCRSP and four phenotypes of individuals with chronic RCRSP. 

The tree selected pain intensity of the previous most painful complaint in addition to shoulder pain as the first outcome to classify individuals with RCRSP. All individuals with chronic RCRSP had no other previous most painful complaints in addition to the shoulder pain with intensity higher than seven points in NPRS, and all individuals with a pain intensity of the previous most painful complaint in addition to the shoulder pain greater than seven points had acute RCRSP. However, pain intensity of the previous most painful complaint in addition to the shoulder pain of seven points or less did not entirely explain the presence of chronic RCRSP by itself. Then, conditioned pain modulation was selected as the second outcome to distinguish between those with acute and chronic RCRSP. Finally, three biological aspects related to shoulder clinical condition and one psychosocial aspect were combined to determine the remaining six biopsychosocial phenotypes. Details about tree divisions with the respective value for cutoff points and the number and percentage of individuals classified in each phenotype are presented in Figure 3.

The three phenotypes identified in those with acute RCRSP (ASP) and the number and percentage of individuals with acute RCRSP in each one were:-ASP-1: presence of a previous painful complaint higher than seven points in another region of the body in addition to shoulder pain (*n* = 4; 26.7%);-ASP-2: presence of a previous pain lower than or equal to seven points and conditioned pain modulation during the cold pressor test higher than 75.8% of change on PPT-deltoid associated with unilateral pain and TSK score higher than 28 points (*n* = 8; 53.3%);-ASP-3: presence of a previous pain lower or equal to seven points and conditioned pain modulation during the cold pressor test lower than or equal to 75.8% of change on PPT-deltoid associated with positive scapular assistance test and angular onset of pain during external rotation lower than or equal to 73 degrees (*n* = 2; 13.3%).

The other four phenotypes were identified in those with chronic RCRSP (CSP) and included individuals with previous pain in another region in addition to the shoulder pain lower than or equal to seven points. In addition to this aspect, the four phenotypes and the number and percentage of individuals with chronic RCRSP in each one were:-CSP-1: conditioned pain modulation during the cold pressor test lower than or equal to 75.8% of change on PPT-deltoid associated with negative scapular assistance test (*n* = 22; 57.9%);-CSP-2: conditioned pain modulation during the cold pressor test lower than or equal to 75.8% of change on PPT-deltoid associated with positive scapular assistance test and angular onset of pain during external rotation higher than 73 degrees (*n* = 8; 21.0%);-CSP-3: conditioned pain modulation during the cold pressor test higher than 75.8% of change on PPT-deltoid associated with bilateral pain (*n* = 6; 15.8%);-CSP-4: conditioned pain modulation during the cold pressor test higher than 75.8% of change on PPT-deltoid associated with unilateral pain and a TSK score lower than or equal to 28 points (*n* = 2; 5.3%).

## 4. Discussion

This study identified three main biopsychosocial phenotypes related to acute RCRSP and four related to chronic RCRSP through a combination of six biopsychosocial aspects. The phenotypes identified to better classify the individuals with acute RCRSP combined the following aspects: ASP-1. presence of a previous painful complaint with intensity higher than seven points in addition to shoulder pain; ASP-2. preserved endogenous modulation of pain associated with unilateral pain and signs of kinesiophobia; and ASP-3. low efficiency of endogenous modulation of pain associated with signs of shoulder pain related to biomechanical alterations. Among individuals with chronic RCRSP, none of the phenotypes showed a pain intensity higher than seven points in another body region. Additionally, the following aspects were combined in phenotypes to better classify individuals with chronic RCRSP: CSP-1. low efficiency of endogenous modulation of pain associated with no signs of shoulder pain related to biomechanical impairments; CSP-2. low efficiency of endogenous modulation of pain associated with signs of shoulder pain related to biomechanical impairments; CSP-3. preserved endogenous modulation of pain associated with bilateral pain; and CSP-4. preserved endogenous modulation of pain associated with unilateral pain and no kinesiophobia.

### 4.1. Interpretation of the Results According to Literature

The current study shows that the first aspect to distinguish individuals with acute RCRSP is the presence of a previous painful complaint with intensity higher than seven points in addition to shoulder pain (phenotype ASP-1). This finding suggests that acute RCRSP may be influenced by another musculoskeletal chronic condition, since patients with chronic pain are more likely to present widespread pain and higher pain intensity [25,96]. In fact, having other concomitant musculoskeletal symptoms reduces the probability of recovery over time in people with shoulder pain [7]. When looking at the functioning of the descending inhibitory system of the four individuals with phenotype ASP-1, they presented CPM during the cold pressor test lower than 75.8% of change, which was the second aspect selected by the tree. Low efficiency of the endogenous descending inhibitory pathways was already described as a relevant aspect of chronic pain conditions [97,98,99,100]. Therefore, this possible state of chronic pain arising from the previous pain might have generated central impairments on pain modulation and could have contributed to the recent development of RCRSP. 

After eliminating another musculoskeletal pain with intensity higher than seven, the cut-off point of 75.8% of change in CPM during the cold pressor test divides the clinical picture into two relevant scenarios: most of those with acute RCRSP (53.3%) showed more than 75.8% of change (phenotype ASP-2), while most of those with chronic RCRSP (78.9%) showed 75.8% or less of change in CPM during the cold pressor test (phenotypes CSP-1 and CSP-2). This is interesting because CPM during the cold pressor test was the only difference observed between individuals with acute and chronic RCRSP. The increase in pain threshold in CPM during the cold pressor test may indicate efficiency of endogenous modulation of pain, which means preserved function of the periaqueductal gray and medullary subnucleus reticularis dorsalis to modulate nociceptive inputs and pain perception through their ascending and descending interactions with other regions of the central nervous system, especially to the dorsolateral horn of the medulla [101,102]. However, a cutoff value that defines functional CPM is unknown. Changes higher than 30% from baseline were considered optimal CPM efficiency in chronic low back pain population [98], and reference values in asymptomatic people range between 30 to 55% of change during the cold pressor test [103,104]. Then, the findings of the present study agree that a robust endogenous modulation of pain may be normally engaged to protect against the development of chronic pain [101,105], even in the presence of any biomechanical alteration. However, it is important to highlight that a simple test paradigm is unlikely to fully reflect the complex neuronal, cognitive, and emotional processes involved in endogenous pain modulation [103].

Despite the preserved endogenous pain modulation, phenotype ASP-2 also showed unilateral pain and signs of kinesiophobia. The classification tree selected a cutoff score of 28 points in TSK score to determine the degree of kinesiophobia, indicating that phenotype ASP-2 shows a certain degree of fear of arm movement or physical activity. Although pain experience might be driven by peripheral nociception in individuals with acute RCRSP [10], kinesiophobia may play an important role in the clinical picture and prognosis of the condition. A TSK baseline score of about 26 points was observed in non-recovered patients with shoulder pain over one year of follow-up [106]. Further, a very low level of evidence suggests that high levels of kinesiophobia seems to predict the persistency of disability overtime [27,36] and it is not clear whether kinesiophobia is a predictor for treatment outcome in shoulder pain [107]. 

Preserved endogenous pain modulation was also found in two less frequent phenotypes among individuals with chronic RCRSP, those with no other pain intensity higher than seven points associated with bilateral RCRSP (phenotype CSP-3) or with unilateral RCRSP without kinesiophobia (phenotype CSP-4). Although apparently controversial, current evidence in chronic pain suggests that a robust endogenous pain inhibition is uncommon but possible [97,98,99,100,108] since low CPM efficiency is not a uniform aspect in individuals with chronic shoulder pain [21,22]. The presence of bilateral symptoms indicates that these individuals might have poor prognosis over time [7], but low levels of kinesiophobia are related to better shoulder prognosis [36], which together may contribute to the robustness of the endogenous pain modulation. Therefore, although uncommon, phenotypes CSP-3 and CSP-4 seem to represent the reality of a subgroup of patients with RCRSP that cope well with their condition. Nevertheless, pain related to biomechanical impairments cannot be excluded and should also be investigated in the future.

Individuals with phenotypes CSP-1 and CSP-2 may represent the great majority of patients with chronic RCRSP seeking treatment. The low efficiency of endogenous pain modulation might represent an imbalance between excitatory and inhibitory sensory inputs indicating that central sensitization drives their pain experience [10], which may be the case in some individuals with RCRSP [109]. Surprisingly, the presence of central sensitization was also found in a minority of individuals with acute RCRSP (13.3%, phenotype ASP-3). However, changes in endogenous modulation may be equally important in acute and chronic pain conditions [10,110] and CPM is improved soon after the noxious stimuli stop [10,97]. For those individuals in which the nervous system modus is unable to normalize sensitivity, a secondary hyperalgesia takes place spreading pain complaints to segmentally unrelated areas [10] and favoring neuroplastic changes that perpetuate central sensitization and result in chronic pain [97]. Although conclusions on the prognostic ability of endogenous pain modulation are still not consistent, less efficient CPM has been shown to predict higher risk for the development of chronic pain [100,105]. Thus, the next step to understand the profile of phenotypes ASP-3, CSP-1 and CSP-2 is checking for pain related to biomechanical alterations at the shoulder complex.

Thereafter, the tree selected two clinical measurements that we recommend to be used in clinical practice to better understand the role of biomechanical alterations of the shoulder complex in RCRSP [3,15,111]: scapular assistance test and angular onset of pain during external rotation. In association with impaired endogenous modulation, individuals with phenotype CSP-1 did not present pain related to scapular motion (negative scapular assistance test), and individuals with phenotypes CSP-2 and ASP-3 presented pain related to decreased scapular posterior tilt (positive scapular assistance test [14,61]) and signs of subacromial impingement. Signs of subacromial impingement were demonstrated with angular onset of pain higher than 63 degrees of external rotation [15]. Changing scapular biomechanics in those with chronic pain and impaired endogenous inhibition (phenotype CSP-1) seems not enough to instantly reduce pain since chronic nociception inhibits motor output [10]. The cut-off point for signs of subacromial impingement is about 63° of external rotation, and therefore, CSP-2 and some individuals with ASP-3 seem to present pain related to subacromial impingement, which might be explained by the decreased corticospinal excitability of the rotator cuff [17]. 

Other biopsychosocial aspects were previously found to be altered in people with shoulder complaints, such as distorted sensory information, health-related quality of life, catastrophizing, worrying, somatization, anxiety and depression [7,8,9,19,28,29,36,106,112]. Although the predictability of some of them was related to shoulder prognosis, they were not different between individuals with acute and chronic RCRSP and none of them were selected by the tree as classifiers of these conditions. This might be because the relationship between duration of the symptoms is not linearly associated with the other predictive aspects. Thus, a complex approach to this relationship, such as the one provided by the classification tree, better represents biopsychosocial phenotypes of this population. Furthermore, age did not differ between individuals with acute and chronic pain, which indicates that age is poorly associated with chronification of RCRSP. However, it is important to highlight that the present study is a cross-sectional study that aimed to provide a multidimensional clinical picture of individuals with RCRSP to help clinicians in assessing the aspects associated with acute and chronic RCRSP, which can support different treatment approaches.

### 4.2. Implications for Clinical Practice

When screening people with acute RCRSP, clinicians should be aware of the presence of a another musculoskeletal condition in a chronic stage that might be related to acute shoulder RCRSP (phenotype ASP-1) and therefore, the therapeutic approach focused on behavioral and psychological strategies may be used for these individuals [113,114]. When no other musculoskeletal pain higher than seven points in addition to the shoulder pain is present, efficiency of endogenous modulation of pain may play the most important role in clinical picture. 

However, the assessment of CPM using PPT can be a difficult task in clinical routine. Other more manageable biopsychosocial aspects can help the clinician to understand the role of the endogenous pain modulation system in clinical settings. When individuals with RCRSP for less than six months present unilateral pain and a TSK score higher than 28 points, endogenous pain modulation may be preserved (phenotype ASP-2). Although in a lower frequency, the same happens when individuals with RCRSP for more than six months present bilateral pain (phenotype CSP-3) or unilateral pain associated with a TSK score lower than 28 points (phenotype CSP-4). In all three phenotypes, peripheral nociceptive input continues to drive the experience of pain, as a primary hyperalgesia [10], and so, they are more likely to present substantial improvements over time or following exercise therapy [10] or shoulder surgery [115,116]. Moreover, as a potential barrier to the practice of exercise and to the recovery of pain conditions [36,117], kinesiophobia might be a relevant psychosocial aspect to be considered by physiotherapy [113,114].

Similarly, for those with RCRSP for more than six months associated with a negative scapular assistance test (phenotype CSP-1) or with a positive scapular assistance test and angular onset of pain higher than 73 degrees of external rotation (phenotype CSP-2), endogenous pain modulation may be impaired. The same happens, in a lower frequency, among those with RCRSP for less than six months, positive scapular assistance test and angular onset of pain lower than 73 degrees of external rotation (phenotype ASP-3). In all three phenotypes, the clinical picture may be dominated by central sensitization [10]. Thus, clinicians should consider that individuals with phenotypes CSP-1 and CSP-2 may benefit from pain neuroscience education strategies and other therapeutic interventions followed by motor control retraining [10]. Those with phenotype ASP-3 may benefit from a biomechanical scapular focused approach associated with pain neuroscience education to avoid persistency of symptoms.

### 4.3. Limitations of the Study

The cut-off point of six months, which was used to differentiate between individuals with acute and chronic RCRSP, might be considered as a limitation of the study, but it is in accordance with the recommendation of the task force on Taxonomy of the International Association for the Study of Pain for research purposes [39]. The suggested time for tissue healing (about three months) does not seem to be sufficient to decide a point of division between acute and chronic pain [39]. Given that intrinsic and extrinsic factors may contribute to the tissue healing process, pain can be associated with actual or potential tissue damage [118]. Findings of bilateral alterations in the rotator cuff and acromioclavicular joint in individuals with atraumatic unilateral shoulder pain support the mismatch between tissue damage and shoulder pain [119]. Moreover, other findings show that neuroplastic changes with abnormal brain chemistry and neuronal loss or dysfunction is related to persistent pain for longer than six months [120]. For all these reasons, the use of six months as a cut-off point is recommended for research purposes [39] and it is probably not a limitation of the study since it ensures that tissue repair, if present, has already taken place in order to label the individual with chronic RCRSP.

Low endogenous inhibition leads to chronic nociception, which inhibits motor output [10]. A limitation of this study was the lack of inclusion of other variables that possibly influence acute and chronic conditions, such as muscle activity and scapular kinematics. Pain perception of patients among different cultures and nationalities can also be influenced by psychosocial aspects [121] so the results of this study cannot be extrapolated to populations of other nationalities. 

Only individuals with shoulder pain for more than four weeks were included to standardize enrollment and ensure stability of the symptomatology; therefore, the results cannot be applied to those with an episode of RCRSP of less than four weeks of duration. Moreover, the assessments were performed during the active phase of the pain, which better translates the study to clinical settings but did not allow conclusions about the origin, persistency of symptoms over time or prognoses. The limited number of participants with acute RCRSP might implicate in the lack of generalizability of the results. However, the leave-one-out cross-validation process applied for the decision tree is a unique method able to guarantee a generalist model for reduced data sets and avoid bias [93]. Decision trees are built to clarify the series of processes that a clinician needs to go through to move a patient from diagnosis to treatment [122]. The current findings are clinically relevant but need further validation with larger samples from other nationalities as well as clinical trials aimed to assess the effect of specific interventions on each phenotype.

## 5. Conclusions

This study identified three biopsychosocial phenotypes related to acute RCRSP and four related to chronic RCRSP. These phenotypes were identified through a complex interaction between six biopsychosocial aspects. Individuals with chronic RCRSP do not have a previous complaint with an intensity higher than seven points in addition to the shoulder pain and seem to have malfunctioning of the endogenous pain modulation system and no signs of shoulder pain related to biomechanical impairments. Individuals with acute RCRSP are more likely to have preserved endogenous modulation of pain but unilateral pain with signs of kinesiophobia. These findings provide evidence that reflects the importance of the biopsychosocial therapeutic approach for individuals with RCRSP from assessment to clinical decision-making and prognosis.

## Figures and Tables

**Figure 1 diagnostics-10-00928-f001:**
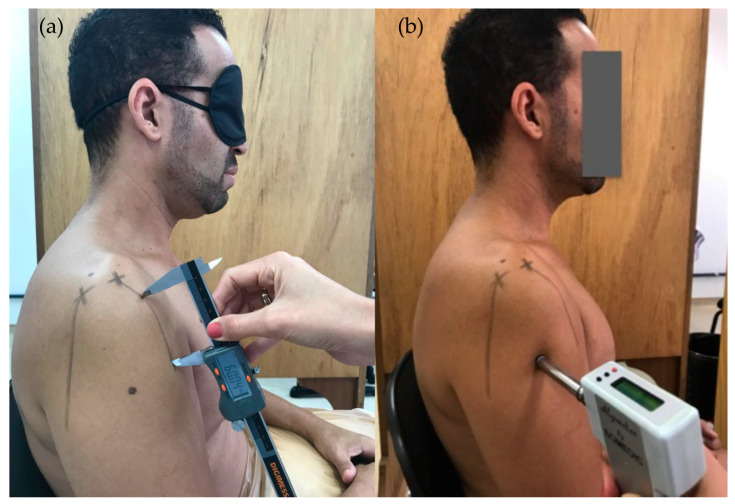
(**a**) Two-point discrimination threshold testing with representation of anterior and posterior lines to guide testing; (**b**) pressure pain threshold testing locally (deltoid muscle) at the shoulder.

**Figure 2 diagnostics-10-00928-f002:**
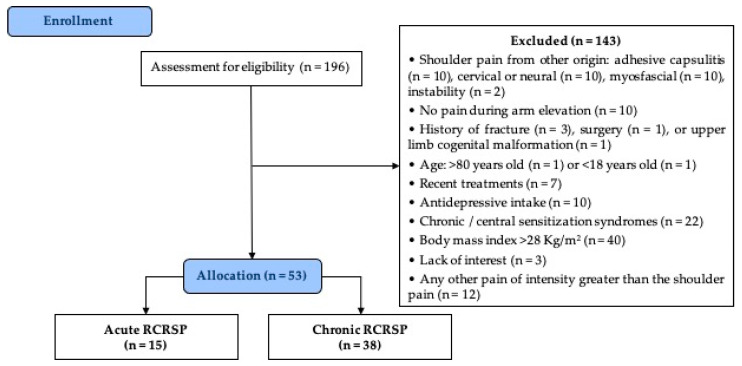
Flow diagram for enrollment and allocation of the participants.

**Figure 3 diagnostics-10-00928-f003:**
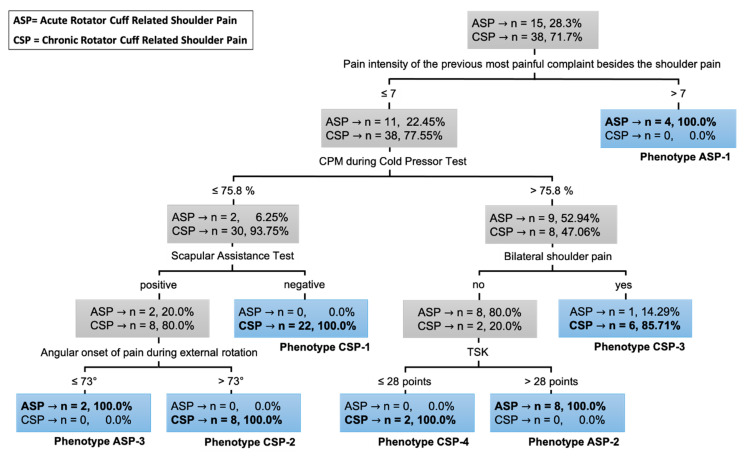
Classification tree model for acute (ASP) and chronic (CSP) rotator cuff related shoulder pain. Abbreviations: CPM—conditioned pain modulation; TSK—Tampa Scale for Kinesiophobia.

**Table 1 diagnostics-10-00928-t001:** Characteristics of the participants related to sociodemographic and clinical outcomes.

	Acute RCRSP (*n* = 15)	Chronic RCRSP (*n* = 38)	*p*-Value
Sociodemographic aspects			
Age (years)	31.0 [20.0]	33.5 [24.3]	0.54
Sex			
Male	9 (60.0%)	23 (60.5%)	0.97
Educational level			
Incomplete elementary school	1 (6.7%)	1 (2.6%)	0.40
Elementary school	0 (0.0)	3 (7.9%)	
High school	6 (40.0%)	9 (23.7%)	
University education	8 (53.3%)	25 (65.8%)	
Marital status			
Single	7 (46.7%)	16 (42.1%)	0.92
Married	7 (46.7%)	18 (47.4%)	
Divorced	1 (6.7%)	3 (7.9%)	
Widowed	0 (0.0%)	1 (2.6%)	
Arm dominance			
Right	6 (40.0%)	9 (23.7%)	0.25
Left	8 (53.3%)	25 (65.8%)	
Employment status			
Active	14 (93.3%)	33 (86.8%)	0.66
Retired	1 (6.7%)	5 (13.2%)	
Occupational demand			
Physical	2 (13.3%)	7 (18.4%)	0.44
Psychological	10 (66.7%)	18 (47.4%)	
Both	3 (20.0%)	13 (34.2%)	
Presence of upper limb repetitive movement			
Yes	11 (73.3%)	25 (65.8%)	0.75
No	4 (26.7%)	13 (34.2%)	
Frequency of physical activity (days/week)	4 [4]	4 [4]	0.73
Smoking status			
Yes	2 (13.3)	5 (13.2)	1.00
No	13 (86.7)	33 (86.8)	
Biological aspects related to general clinical health status			
Presence of any other pain previous to shoulder pain	10 (66.7%)	22 (57.9%)	0.55
Pain intensity of the previous most painful complaint besides the shoulder pain (0–10)	4.8 ± 3.1	2.9 ± 1.6	0.10
CSI-part A	23.0 ± 9.8	28.0 ± 11.3	0.14
Biological aspects related to shoulder clinical condition			
Affected shoulder			
Dominant	6 (40.0%)	14 (36.8%)	0.44
Non-dominant	6 (40.0%)	10 (26.3%)	
Both	3 (20.0%)	14 (36.8%)	
Duration of symptoms (months)	3.0 [2.3]	36.0 [48.8]	0.001
Pain intensity during arm movement (0–10)	8.0 [4.0]	7.0 [2.3]	0.418
ROM (degrees)			
Angular onset of pain during abduction	148.0 [35]	132.0 [64.0]	0.065
Angular offset of pain during abduction	175.0 [14]	169.0 [15.0]	0.43
Angular onset of pain during external rotation	85.5 ± 15.3	92.1 ± 19.0	0.23
Angular offset of pain during external rotation	95.9 ± 11.9	99.0 ± 13.1	0.43
Scapular dyskinesis			
Present	14 (93.3%)	36 (94.7%)	1.00
Absent	1 (6.7%)	2 (5.3%)	
Scapular assistance test			
Positive	5 (33.3%)	11 (28.9%)	0.75
Negative	10 (66.7%)	27 (71.1%)	
Total number of positive special tests for shoulder rotator cuff related pain	4.5 ± 2.1	4.9 ± 2.0	0.46
DASH	12.5 [15.0]	18.3 [16.6]	0.295

Data are mean ± standard deviation, median (IQT range), or frequency (%). Abbreviations: ROM—range of motion; CSI—Central Sensitization Inventory; DASH—Disabilities of the Arm, Shoulder and Hand questionnaire.

**Table 2 diagnostics-10-00928-t002:** Characteristics of the participants related to sensory and psychosocial outcomes.

	Acute RCRSP (*n* = 15)	Chronic RCRSP (*n* = 38)	*p*-Value
Biological aspects related to sensory function			
TPDT-anterior (mm)	34.9 ± 19.0	40.2 ± 15.8	0.30
TPDT-posterior (mm)	44.1 ± 16.7	44.9 ± 12.6	0.85
LRJT-accuracy (%)	100.0 [10.0]	100.0 [0.0]	0.55
LRJT-time (s)	1.4 [0.4]	1.3 [0.5]	0.60
PPT-acromion (KPa)	305.6 [215.0]	271.0 [255.5]	0.79
PPT-deltoid (KPa)	353.4 ± 163.3	360.0 ± 173.8	0.90
PPT-tibialis anterior (KPa)	379.0 [160.3]	376.8 [208.4]	0.91
TS-acromion (0–10)	3.0 [3.0]	2.0 [3.0]	0.62
TS-tibialis anterior (0–10)	3.0 [2.0]	2.0 [2.3]	0.15
CPM during Cold Pressor Test (% change)	82.2 ± 44.7	49.8 ± 42.5	0.02 *
CPM post-Cold Pressor Test (% change)	7.2 ± 28.9	11.8 ± 32.5	0.64
Psychosocial aspects			
FABQ-Br			
FABQ-PA	15.1 ± 4.7	11.8 ± 6.4	0.08
FABQ-W	7.0 [13.0]	8.5 [15.8]	0.70
TSK	37.1 ± 3.2	35.9 ± 7.2	0.55
PCS			
Rumination	6.7 ± 2.6	6.4 ± 3.4	0.80
Magnification	4.0 [5.0]	3.0 [3.5]	0.95
Helplessness	5.0 [10.0]	3.0 [6.3]	0.46
CPSS			
Pain management	400.0 [110.0]	390.0 [115.0]	0.48
Coping with symptoms	612.0 ± 117.0	608.2 ± 109.2	0.91
Physical function	880.0 [110.0]	875.0 [95.0]	0.98
Total score	1950.0 [350.0]	1860. [330.0]	0.68
DASS-21			
Depression	0.0 [4.0]	2.0 [3.0]	0.43
Anxiety	1.0 [2.0]	1.0 [2.3]	0.41
Stress	1.0 [5.0]	3.5 [7.3]	0.47
EQ-5 D	0.82 [0.15]	0.88 [0.11]	0.93

Abbreviations: TPDT—two-point discrimination threshold; LRJT—left/right judgment task; PPT—pressure pain threshold; TS—temporal summation; CPM—conditioned pain modulation; TSK—Tampa Scale for Kinesiophobia; PCS—Pain Catastrophizing Scale; CPSS—Chronic Pain Self-Efficacy Scale; DASS-21—Depression, Anxiety and Stress Scale-21; EQ-5D—EuroQoL instrument; * *p* < 0.05 represents significant difference between groups for CPM during the cold pressor test.

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
