# Peer review of "Biopsychosocial Aspects in Individuals with Acute and Chronic Rotator Cuff Related Shoulder Pain: Classification Based on a Decision Tree Analysis"

_diagnostics, 2020, doi:10.3390/diagnostics10110928_

Round 1
Reviewer 1 Report
I think this is a very pertinent and interesting research topic since rotator cuff related shoulder pain present an elevate prevalence. In addition, the present study deepens in an important pain aspect such as biopsychosocial characteristics of patients that has been subject of interest in recent years.
For me, one of the most important limitation of the present study is the scarce number of study subject, limiting the extrapolation of the results obtained beyond the study sample. I encourage the authors to continue this interesting research line in other study populations with higher number of participants, that allow to reach more solid study conclusion in this interesting issue.
However, in my opinion some aspect of the present study should be improved:
- An aspect that should be interesting to know will be the location, country, the region, the environment (hospital, university…) where the data were collected.
- Please, provide in text more information about the selection process of participants. How have authors reached the final fifty-three patients enrolled in the study?
- Should be interesting to provide some information about sample size calculation. Why have authors considered that 53 patients were enough to reach solid conclusions?
- Between lines 126-128, authors have mentioned that they have employ the Numeric Pain Rating Scale (NPRS) to evaluate “the presence of any other pain previous to shoulder pain and the pain intensity of the previous most painful complaint besides the shoulder pain”. Authors must provide the appropriate reference for this scale and some reliability data that allow reader to understand the reliability level of the scale employed.
- In section “3 Sociodemographic outcomes”, authors must describe the criteria they have followed to stablish that a participant is smoker or non smoker. In the same mode, author should explain the criteria followed to quantify the weekly frequency of physical activity, how has it been measured? in days? In hours? This issue should be also corrected in table 1, for a better readers understanding.
- In line 104, authors explain that “The study was approved by the University Human Research Ethics Committee (No. XX)”, but which University has approved the study through his Ethics Committee? This information should be described correctly, providing the name of the university that has approved the study through his Research Ethics Committee.
- Please, authors must provide the code of approval of the Ethics Committee.
- I recommend that in section “7 Psychosocial outcomes”, some reliability data of each questionnaire employed should be provided to facilitate reader to understand the reliability of the questionnaire employed.
Author Response
Response to Editor’s comments
Please revise the manuscript according to the reviewers' comments and upload the revised file within 10 days. Use the version of your manuscript found at the above link for your revisions, as the editorial office may have made formatting changes to your original submission. Any revisions should be clearly highlighted, for example using the "Track Changes" function in Microsoft Word, so that changes are easily visible to the editors and reviewers. Please provide a cover letter to explain point-by-point the details of the revisions in the manuscript and your responses to the reviewers' comments. Please include in your rebuttal if you found it impossible to address certain comments. The revised version will be inspected by the editors and reviewers. Please detail the revisions that have been made, citing the line number and exact change, so that the editor can check the changes expeditiously. Simple statements like ‘done’ or ‘revised as requested’ will not be accepted unless the change is simply a typographical error.
Response: We appreciate the opportunity to re-submit this manuscript. Thank you for your time and effort in reviewing this manuscript. We appreciate all your comments and recommendations that led to great improvements in this study. All changes are highlighted in light gray. We expect to attend your concerns and that you reconsider this study for publication based on the replies below.
Response to Reviewer 1 Comments
I think this is a very pertinent and interesting research topic since rotator cuff related shoulder pain present an elevate prevalence. In addition, the present study deepens in an important pain aspect such as biopsychosocial characteristics of patients that has been subject of interest in recent years.
For me, one of the most important limitation of the present study is the scarce number of study subject, limiting the extrapolation of the results obtained beyond the study sample. I encourage the authors to continue this interesting research line in other study populations with higher number of participants, that allow to reach more solid study conclusion in this interesting issue.
Response: Thanks for your careful revision. We agree that the low number of subjects might be a limitation of the generalizability of the results of this study. However, we believe this study brings important and new data about rotator cuff related shoulder pain and may provide insights for further studies and add to clinical practice. Our research team will surely continue in this interesting research line. We have added the lack of generalizability of the results as a limitation of the study (page XXX, lines XXX).
“The limited number of participants with acute RCRSP might implicate in the lack of generalizability of the results.”
However, in my opinion some aspect of the present study should be improved:
Point 1: An aspect that should be interesting to know will be the location, country, the region, the environment (hospital, university…) where the data were collected.
Response: This information was added on page xxx (lines xxx).
Point 2: Please, provide in text more information about the selection process of participants. How have authors reached the final fifty-three patients enrolled in the study?
Response: More details about the enrollment and selection processes are presented in Figure 2 and on page xxx (lines xxx):
“One hundred ninety-six potential individuals were initially recruited. One hundred forty-three individuals were excluded, and reasons are presented in the flowchart of the study (Figure 2).”
Point 3: Should be interesting to provide some information about sample size calculation. Why have authors considered that 53 patients were enough to reach solid conclusions?
Rushing, C.; Bulusu, A.; Hurwitz, H.I.; Nixon, A.B.; Pang, H. A leave-one-out cross-validation SAS macro for the identification of markers associated with survival. Comput. Biol. Med. 2015, 57, 123–129.
Jayawardana, K.; Schramm, S.-J.; Tembe, V.; Mueller, S.; Thompson, J.F.; Scolyer, R.A.; Mann, G.J.; Yang, J. Identification, Review, and Systematic Cross-Validation of microRNA Prognostic Signatures in Metastatic Melanoma. J. Invest. Dermatol. 2016, 136, 245–254
Point 4: Between lines 126-128, authors have mentioned that they have employ the Numeric Pain Rating Scale (NPRS) to evaluate “the presence of any other pain previous to shoulder pain and the pain intensity of the previous most painful complaint besides the shoulder pain”. Authors must provide the appropriate reference for this scale and some reliability data that allow reader to understand the reliability level of the scale employed.
Response: Thanks for your comment. Reliability data for individuals with musculoskeletal pain was provided for the use of the scale (page 3, lines XX-XX):
“The 11-point scale provide sufficient levels of discrimination, in general, to describe pain intensity in individuals with pain [51]with ICC ranging from 0.93 to 0.99 in individuals with musculoskeletal pain [52,53].”
Point 5: In section “3 Sociodemographic outcomes”, authors must describe the criteria they have followed to stablish that a participant is smoker or non-smoker. In the same mode, author should explain the criteria followed to quantify the weekly frequency of physical activity, how has it been measured? in days? In hours? This issue should be also corrected in table 1, for a better readers understanding.
Response: Participant was considered as a smoker or non-smoker based on a yes or no question (Do you smoke?). Frequency of physical activity was based on how many days a week. This information was added on page xxx (lines xxx). The unit for physical activity measurement was also added in Table 1.
Point 6: In line 104, authors explain that “The study was approved by the University Human Research Ethics Committee (No. XX)”, but which University has approved the study through his Ethics Committee? This information should be described correctly, providing the name of the university that has approved the study through his Research Ethics Committee.
Response: The information was added on page 3 (lines XX-XX).
“The study was approved by the Human Research Ethics Committee of the Federal University of São Carlos (CAAE 71447317.6.0000.5504) and conducted according to the Declaration of Helsinki. All participants gave written informed consent prior to their enrollment in the study.”
Point 7: Please, authors must provide the code of approval of the Ethics Committee.
Response: The information was added on page 3 (lines XX-XX).
“The study was approved by the Human Research Ethics Committee of the Federal University of São Carlos (CAAE 71447317.6.0000.5504) and conducted according to the Declaration of Helsinki. All participants gave written informed consent prior to their enrollment in the study.”
Point 8: I recommend that in section “7 Psychosocial outcomes”, some reliability data of each questionnaire employed should be provided to facilitate reader to understand the reliability of the questionnaire employed.
Response: Thanks for your suggestion. Reliability data were included for all the questionnaires on pages 4 (lines 150-151) and 7 (lines 288-289; X-XX).
Lines 150-151: “Reliability of CSI showed Cronbach’s alfa of 0.91 [56].”
Lines 288-289: “Test-retest reliability showed Intraclass Correlation Coefficient (ICC) of 0.94 for FABQ-PA, 0.82 for FABQ-W and 0.82 for TSK [85].”
Line 299: “Test-retest reliability of this version of PCS showed ICC of 0.88 [88].”
Lines 304-305: “Internal consistency of the scale showed Cronbach’s alfa of 0.94 for all items [89].”
Lines 308-310: “The Brazilian version of DASS-21 is reliable and valid [90]. Internal consistency of the scale showed Cronbach’s alfa of 0.92 for depression, 0.90 for stress, and 0.86 for anxiety [90].”
Line 315: “Test-retest reliability of the EQ-5D showed ICC of 0.58 to 0.89 [92].”

Reviewer 2 Report
This is a generally well-written manuscript, with a highly relevant research topic. It assess the opportunity of an alternative approach to capture the nonlinear relationships between multiple biopsychosocial variables in persons with RCRSP.
Introduction:
L79: The classification and regression tree may be an alternative approach to capture the nonlinear relationships between multiple variables and produce results that can be easily applied in clinical practice.
It is not clear for the reader in which way these kind of analysis provide more or other information which can be easily applied. What is the difference in outcome between analyses methods that is so important for clinicians? From what I read in the result section (L342) this is not clear to me how it provides important information for clinicians
L85: The aim of this study was to determine which biopsychosocial aspects would better classify individuals with acute and chronic RCRSP and describe how those aspects interact among these two phenotypes of pain.
The aim is not completely clear for me. What is meant with ‘those aspects’? Furthermore, the term phenotypes is not introduced, and from the described aim, I think persons with acute versus persons with chronic pain are meant.
This means that a phenotype is here defined according to duration of complaints? This should be clarified in the introduction, so the aims are logical. (see also comment in general questions)
Methods:
- Some in-exclusion criteria are not clear to me, together with how these are assessed. It was only by reading Table 1 that some details became clear to me (which is referred to in the result section)
Why are persons with a previous diagnosis of central sensitization excluded, when you assess pain mechanisms/sensory function as well? Don’t you create a selection bias when persons with central sensitization are excluded? How was this assessed? I see 22 persons were excluded based on this criterium. Also pain in other region higher (n=12 exclusion): why are these excluded?
Why were persons excluded who received treatment?
In addition, the following inclusion criteria seems to confirm the aim to include persons with dominant nociceptive pain: Inclusion criteria were pain over the deltoid and/or upper arm region for more than 4 weeks, pain associated to arm movement, and familiar pain reproduced with loading or resisted testing in abduction or external rotation
- L124 presence of any kind of hypersensitivity and presence of any other frequent symptoms (fatigue, concentration difficulties, sleep disturbance, swollen feeling, tingling, or numbness): : how were these aspects assessed?
-The presence of scapular dyskinesis was assessed according to literature: I think it is better to specifically describe the used method and the possible outcomes. The same holds for the scapular assistance test: What is the outcome of this test.
-L144: Regarding the special tests / orthopedic tests assessed: Is it not problematic to use non-valid tests in the analysis? It is clear that in RCRPS, most of these test are positive. The use of ‘structural based’ tests in persons diagnosed with ‘a syndrome’ (not specifically structural-related) seems contradictory…
-L148: For that, I think a clear reference is needed
-L 165: TPDT , PPTs, TS, CPM: What are the exact outcomes of these tests used in the analyses?
-L243: Why both the FABQ and TSK are used? They generally measure the same construct. However, the have their own specific information, but it is unclear at this moment whether this distinct difference between the assessed constructs is essential for this study’s purpose. This should be clarified.
- The CSI is a self-report measure of symptoms related to central sensitization, not a measure of psychosocial outcomes; So it is not correct to place it under the subheading psychosocial outcomes.
- I think that a table which provides an overview of the different methods used, together with their specific outcomes (continuous with range; ordinal), would be of interest to include.
- It is not clear to me why both analysis (group differences; creating algorithms) are performed? What is there additional benefit?
Results
L341: CPM-during?
Discussion
The discussion is difficult to follow. It is suggested to add subtitles to increase the readability/comprehensiveness. For example, result interpretation according to literature, implications for clinical practice (regarding assessment and intervention), future research, limitations (specifying influence of in-exclusion criteria on study results, maybe limitations of the data-analysis method used, …)
- L 383: The decision tree identified three main biopsychosocial phenotypes related to acute 383 RCRSP and four related to chronic RCRSP through a combination of six biopsychosocial aspects.
Were in the text can the reader clearly find this information? Which 6 BPS aspects? Which variables are included in the 3 BPS phenotypes in Acute RCRSP and which in the 4 phenotypes in chronic RCRPS? This is not clear from the text under 3.3. (L342)
L 387—396: better to add in the result section?
- L 386: this is not clear to me (see comment in general questions)
- L 399-401: this does not supports the exclusion criteria of a previous diagnosis of central sensitization.
- L 428: which 2 phenotypes?
- 436: Classification tree selected a cutoff score of 28 points in TSK score to determine the degree of kinesiophobia à how can this be interpreted in context of the cut-off of 37 points on the tsk which is known from low back pain to distinguish between persons with low versus high pain-related fear of movement?
It is suggested to furthermore avoid the term kinesiophobia, since ‘phobia’ might suggest a psychiatric disease, which is definitely not the case for persons with higher levels of pain-related fear of movement.
- 347-348: indicating that they are unable to cope with their pain symptoms. Do I read it correctly that persons who score higher than 28 points on the tsk are referred to as unable to cope with their pain symptoms? This interpretation of the TSK is not correct, and should be altered.
442: this contrast to the results found in persons with shoulder pain receiving physiotherapy (cfr https://pubmed.ncbi.nlm.nih.gov/30791696/). At this moment it seems not clear that pain-related fear of movement is a predictor for treatment outcome in persons with shoulder pain in general.
- L 458: fear of pain? Based on which PROM?
General questions:
- I do not think the term ‘acute’ is appropriate. First of all, persons are only included when they experience shoulder pain for more than 4 wks. Furthermore, the acute group includes persons which have pain up to six months. That is no acute pain any longer.
Another term, describing the difference in groups based on duration of complaints is needed.
- Why do you define 2 phenotypes, and don’t add duration of complaints as a variable in the analyses?
In the statistical analyses section, I read that the group is divided based on duration of pain: can you call this a phenotype?(cfr aim in introduction) Is a ‘phenotype’ not the result of a statistical analysis? It is also written as such in the discussion on line 383: The decision tree identified three main biopsychosocial phenotypes related to acute 383 RCRSP and four related to chronic RCRSP
Why 6 months? This seems an arbitrary point… Regarding to the phases of tissue healing, important aspects related to recovery seems to happen earlier, around, 12 weeks following pain onset. Why not identifying that subgroup of persons in the early stage? When tissue is healing, inflammatory pain might be present first, which is typically more irritable; higher in intensity (augmented nociceptive pain). So I think that the range of 6 months is to wide and does include different clinical phases, and thus subgroups… Potentially for this reason, no differences are found between groups…
L 386: which provides evidence for the complexity of RCRSP and for the presence of subgroups (or 386 phenotypes) of individuals with acute and chronic RCRSP: do you mean different subgroups within the acute and within the chronic pain group? Cfr what I stated above?
Furthermore, The fact that analyses are on unequal groups, it that a problem?
Author Response
Please, see the attachment.

Reviewer 3 Report
Dear Author(s),
The interesting study and theme, but not well presented. As an orthopaedic suregeon I want to know clear and exact facts - what factors should I search for in my outpatient practice.
From the text it is not clear what is acute and what chronic rotator cuff related pain. Because from statement; "...94 criteria were pain over the deltoid and/or upper arm region for more than 4 weeks,.." - if we conclude the pain lasted for 4 weeks as inclusion criteria, that cannot be acute pain ?!
Conclusions ar not clearly defined.
I think the paper has a scientific potential, but the conclusions are not simple, clear and easy to remembered and used in practice.
Best regards,
Reviewer
Author Response
Please, see the attachment.

Round 2
Reviewer 2 Report
Thank you for answer all the comments. I do not have any further comments